

# Characterization of genes encoding small heat shock proteins from *Bemisia tabaci* and expression under thermal stress

Jing Bai[1], Xiao-Na Liu[1], Ming-Xing Lu[1] and Yu-Zhou Du[1,2]

[1] College of Horticulture and Plant Protection & Institute of Applied Entomology, Yangzhou University, Yangzhou, China
[2] Joint International Research Laboratory of Agriculture and Agri-Product Safety, the Ministry of Education, Yangzhou University, Yangzhou, Jiangsu, China

## ABSTRACT

Small heat shock proteins (sHSPs) are probably the most diverse in structure and function among the various super-families of stress proteins, and they play essential roles in various biological processes. The sweet potato whitefly, *Bemisia tabaci* (Gennadius), feeds in the phloem, transmits several plant viruses, and is an important pest on cotton, vegetables and ornamentals. In this research, we isolated and characterized three α-crystallin/sHSP family genes (*Bthsp19.5*, *Bthsp19.2*, and *Bthsp21.3*) from *Bemisia tabaci*. The three cDNAs encoded proteins of 171, 169, and 189 amino acids with calculated molecular weights of 19.5, 19.2, and 21.3 kDa and isoelectric points of 6.1, 6.2, and 6.0, respectively. The deduced amino acid sequences of the three genes showed strong similarity to sHSPs identified in Hemiptera and Thysanoptera insects species. All three sHSPs genes from *Bemisia tabaci* lacked introns. Quantitative real-time PCR analyses revealed that the three *Bt*HSPs genes were significantly up-regulated in *Bemisia tabaci* adults and pupae during high temperature stress (39, 41, 43, and 45 °C) but not in response to cold temperature stress (−6, −8, −10, and −12 °C). The expression levels of *Bthsp19.2* and *Bthsp21.3* in pupae was higher than adults in response to heat stress, while the expression level of *Bthsp19.5* in adults was higher than pupae. In conclusion, this research results show that the sHSP genes of *Bemisia tabaci* had shown differential expression changes under thermal stress.

Corresponding author
Yu-Zhou Du, yzdu@yzu.edu.cn

## INTRODUCTION

Heat shock proteins (HSPs) comprise a group of highly-conserved proteins that are widely found in prokaryotes and eukaryotes. At present, HSPs can be divided into HSP100, HSP90, HSP70, HSP60, HSP40 and small heat shock proteins (sHSPs) according to their molecular weight and homology (*Kim, Kim & Kim, 1998*; *Sørensen, Kristensen & Loeschcke, 2003*). sHSPs are the least conserved family of all HSPs and were first identified as a set of low molecular proteins (15–43 kDa) induced after heat shock in *Drosophila melanogaster* (*Tissières, Mitchell & Tracy, 1974*). sHSPs exhibit more diversity in sequence, structure, size, and function compared to other types of HSPs (*De Jong, Caspers & Leunissen, 1998*;

*Franck et al., 2004*). They are a superfamily of proteins that contain an α-crystallin domain and variable N- and C-terminal extensions (*Stromer et al., 2004*). Because the members of this protein family are very dynamic and heterogeneous, information on structure and function is lacking (*Haslbeck et al., 2005*; *Horwitz, 2003*). Some sHSPs act as molecular chaperones to block the aggregation of unfolded proteins and have properties of protecting cell under stress environment (*Garrido et al., 2012*). sHSPs also play some important roles in apoptosis and autophagy, actin and intermediate filament kinetics, cytoskeletal tissue and membrane fluidity in addition to the stress response (*Haslbeck, 2002*; *Quinlan, 2002*; *Tsvetkova et al., 2002*; *Sun & MacRae, 2005*). Since insects have a strong ability to adapt to a variety of habitats, it is important to understand the roles of sHSP in invertebrates. According to previous studies, sHSPs play important roles in adaptation to hot/cold stress, metamorphosis, normal development, diapause, and the immune response (*Parsell & Lindquist, 1993*; *Jakob & Buchner, 1994*; *Feder & Hofmann, 1999*; *Hayward et al., 2005*; *Song et al., 2006*; *Huang & Kang, 2007*; *Rinehart et al., 2007*; *Gu et al., 2012*; *Lu et al., 2014*; *Pan et al., 2017*).

*Bemisia tabaci* (Gennadius), also known as sweet potato whitefly, belongs to the order Hemiptera and is an important pest of cotton, vegetables, and bedding plants. It causes damage by ingesting phloem, transmitting plant viruses, and also by secreting honeydew, which can decrease photosynthesis. *Bemisia tabaci* has caused large economic losses in agricultural production worldwide (*De Barro et al., 2011*). As one of the 100 most invasive organisms in the world, the occurrence of *Bemisia tabaci* is also very serious in China. In particular, the Mediterranean (MED) or Q biotype of *Bemisia tabaci* has been rapidly expanding and causing harm in China with increased fecundity, a shorter developmental period and stronger resistance to stress since its first discovery in 2005 (*Chu et al., 2005*; *Xu, Wang & Liu, 2006*). Therefore, it is very necessary to conduct relevant research on this insect.

It is well known that temperature is one of the important factors affecting the survival and growth of insects. Many HSP genes including sHSP genes related to temperature adaption and stress have been screened and identified in different insect species (*Hoffmann, Sørensen & Loeschcke, 2003*; *McMillan et al., 2005*; *Somero, 2005*; *Nann, Myriam & Patricia, 2006*; *Pan et al., 2017, 2018*; *Qin et al., 2018*; *Yu, Lu & Cui, 2018*; *Xiong et al., 2018*). Although the sHSP of *Bemisia tabaci* (heat shock protein 20, *hsp20*) has been studied before, there are still some deficiencies in this research scopes. So in this paper, three sHSP genes were cloned and analyzed, and expression patterns were analyzed in detail, further enriching our knowledge of *Bemisia tabaci* HSPs.

## MATERIALS AND METHODS

### Insects

The *Bemisia tabaci* populations used in this study were collected from vegetable farms in Yangzhou (Jiangsu, China). MED cryptic species was determined by using mtDNA COI gene. Insects were reared in environmental chambers on tomato plants at 26 ± 1 °C, with a 16L: 8D photoperiod and 60% relative humidity.

**Table 1  Primer sequences used in RACE and real-time quantitative PCR.**

| Gene | Primer sequence (5′ → 3′) |
| --- | --- |
| RACE | |
| *hsp19.5* | |
| 5′ | TTGGCAGAAGGTAACGGCGGGTGA |
| 3′ | CCAGTCACCAAAACCAACGCCCCA |
| *hsp19.2* | |
| 5′ | CGTTCTTCGTGTTTGGCGTGGAT |
| 3′ | CCCGCAATCAAACAGGAACAAGC |
| *hsp21.3* | |
| 5′ | CTTCCAACAAGTAGGGCAAGAGAGAC |
| 3′ | AACCAATGCTCCCGCAATCAAACAGG |
| Real time PCR | |
| *hsp19.5* | TGAGGAGCGTAGTGATGAAC |
| | CCTTATCGTTGGTGATTGCC |
| *hsp19.2* | GCCAAACACGAAGAACGCAG |
| | CTTGAAGTCAAGGCTTCCGC |
| *hsp21.3* | GCCAAACACGAAGAACGCAG |
| | CAAGGCTTCCGCATTGACGT |
| *EF-1α* | TAGCCTTGTGCCAATTTCCG |
| | CCTTCAGCATTACCGTCC |

## Temperature treatments

Cohorts of 120 *Bemisia tabaci* adults and pupae were collected respectively, placed in glass tubes, and exposed to each temperature treatments (−12, −10, −8, −6, 39, 41, 43, and 45 °C) for 1 h in a constant-temperature subzero incubator (DC-3010; Jiangnan Equipment, Ningbo, China). Treated adults and pupae were allowed to recover at 25 °C for 1 h and were then frozen in liquid nitrogen and stored at −80 °C. Adults and pupae exposed to 26 °C were included as a control. Each treatment included four biological replications ($N = 4$).

## Total RNA, cloning and RACE

Total RNA was extracted from *Bemisia tabaci* adults and pupae using the SV Total RNA Isolation and Purification Kit (Promega, Madison, WI, USA). The integrity, purity and concentration of RNA were examined using 1% agarose gel electrophoresis and spectrophotometry at 260 and 280 nm (Eppendorf BioPhotometer plus, Eppendorf, Germany). RNA samples were stored at −80 °C until needed. cDNA was synthesized using an oligo(dT)$_{18}$ primer (TaKaRa, Dalian, China), and full-length cDNAs of genes encoding sHSPs were obtained by 5′-and 3′-RACE (SMART RACE, Clontech, Mountain View, CA, USA) using the primers shown in Table 1. Full-length sequences were confirmed by RACE 5′ cDNA.

## Characterization genome of *Bemisia tabaci*

The genomic DNA of *Bemisia tabaci* was extracted as described previously (*Xu et al., 2014*). Pairs of specific primers (Table 1) were designed to amplify genomic fragments from the three *Bt*sHSP genes. Products were purified using a gel extraction kit (Axygen, Union City,

CA, USA), cloned into PGEM-T Easy vector (Promega, Madison, WI, USA), and transformed into competent *Escherichia coli* DH5α cells for sequencing.

Open reading frames (ORFs) were identified using ORF Finder software (http://www.ncbi.nlm.nih.gov/gorf/gorf.html). The deduced amino acid sequences were aligned using ClustalX software (version 1.83) (*Thompson, Gibson & Higgins, 2002*). Sequence analysis tools of the ExPASy Molecular Biology Server of the Swiss Institute of Bioinformatics, including Translate, Compute pI/MW, and Blast, were used to analyze the deduced small HSP sequences. Amino acid sequences were used to estimate phylogeny using neighbor-joining, minimum evolution, maximum likelihood, and maximum parsimony methods. Phylogenetic trees were constructed with 1,000 bootstrap replicates using MEGA (version 7.0) (*Kumar, Stecher & Tamura, 2016*).

**Quantitative real-time PCR**

Total RNA was extracted using the SV Total RNA isolation system (Z3100; Promega, Madison, WI, USA). The integrity of RNA in all samples was verified by comparing ribosomal RNA bands in ethidium bromide-stained gels. RNA purity was evaluated by spectrophotometry (Eppendorf BioPhotometer plus) at 260 and 280 nm. Following Quantitative real-time PCR (qPCR), the homogeneity of PCR products was confirmed by melting curve analysis, which was read every 5 s per 0.5 °C increment from 65 to 95 °C. Each reaction was performed in triplicate, and the means of three independent replicates were calculated. qPCR reactions were performed in 20 μl total reaction volumes comprised of 10 μl of 2×SYBR® Premix EXTaq™ (TaKaRa, Dalian, China) master mix, one μl of each gene specific primer (Table 1), and two μl of cDNA templates. It was carried out that reactions on a CFX-Connect real-time PCR system (Bio-Rad, Hercules, CA, USA) using the following conditions: 3 min at 95 °C, 40 cycles of denaturation at 95 °C for 30 s, and annealing (30 s) at the Tm for each gene. The quantity of *Bt*sHSP mRNA was calculated using the $2^{-\Delta\Delta Ct}$ method and normalized to the abundance of the gene encoding elongation factor 1 α (*EF-1α*) (*Li et al., 2013*).

**Data analysis**

We used Levene's test to evaluate the homogeneity of variances among different groups. Significant differences between treatments were identified with either an LSD test (homogeneity of variances) or Dunnett's C test (nonhomogeneous) for multiple comparisons. The data were analyzed using SPSS 16.0 software (*Pallant, 2005*) and represented as means ± standard error.

## RESULTS

**Sequence analysis of *Bemisia tabaci* sHSPs**

Three *Bemisia tabaci* sHSPs were identified and designated *Bthsp19.5*, *Bthsp19.2*, and *Bthsp21.3*; these were deposited in GenBank as accession nos. MF114301, MF114302 and MF114303, respectively. Comparing with the databases of GenBank and PROSITE, the three deduced proteins had high similarity to sHSP family and contained the typical α-crystalline domain at the following locations: *Bt*HSP19.5, 46–155 aa (Fig. S1A);

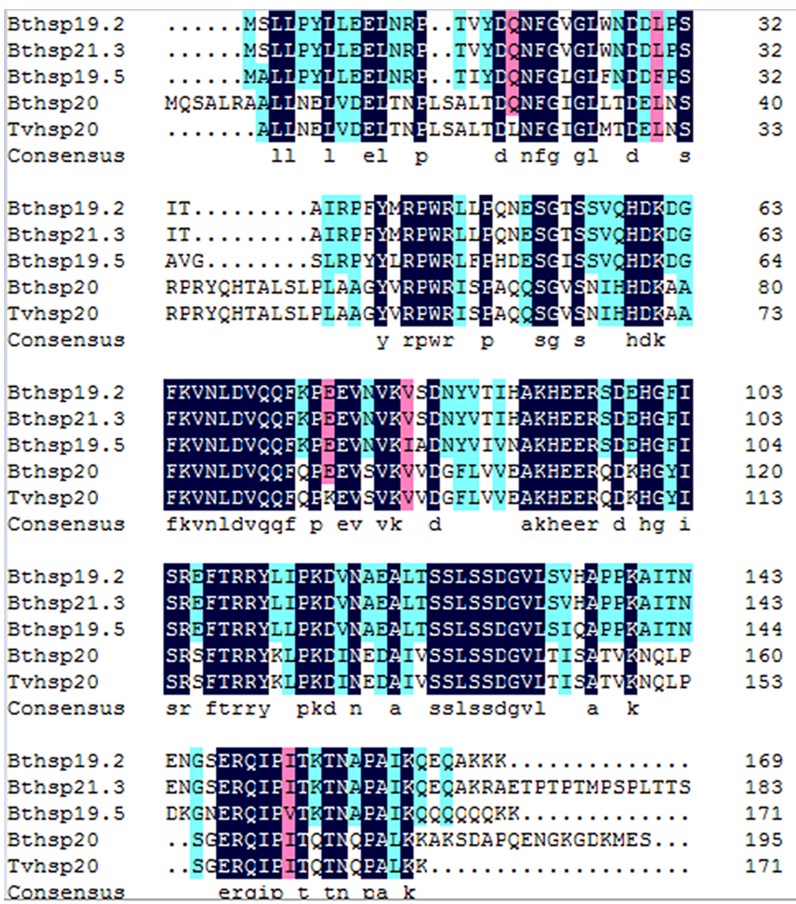

**Figure 1 Alignment of the deduced amino acid sequences.** Alignment of the deduced amino acid sequences of *Bt*HSP19.2, *Bt*HSP21.3, *Bt*HSP19.5, *Bt*HSP20, and *Tv*HSP20. Abbreviations: *Bt*, *Bemisia tabaci*; *Tv*, *Trialeurodes vaporariorum*.

*Bt*HSP19.2, 45–154 aa (Fig. S1B); and *Bt*HSP21.3, 45–154 aa (Fig. S1C). The three full-length cDNAs were 1,225, 1,107, and 1,209 bp, respectively. The deduced protein products contained 171, 169, and 189 amino acids with isoelectric points of 6.1, 6.2, and 6.0, respectively. Each 3′ UTR contained a polyadenylation signal (AATAAA) located 13 (*Bthsp19.5*), 16 (*Bthsp19.2*), and 32 bp (*Bthsp21.3*) upstream of the poly(A) tract. Multiple sequence alignment (Fig. 1) revealed high identity between the three *Bt*HSPs.

## Phylogenetic analysis of *Bemisia tabaci* sHSPs

The three sHSPs' deduced amino acid sequences were compared with orthologous proteins that reported in other insects (Table 2). The ClustalX software revealed *Bemisia tabaci* HSP19.5, HSP19.2, and HSP21.3 showed high sequence identity with HSP20 in *Bemisia tabaci* and *Frankliniella occidentalis* (Fig. 1). The MEGA revealed four phylogenetic trees (neighbor-joining, minimum evolution, maximum likelihood, and maximum parsimony methods) with a similar trend, so we choose neighbor-joining phylogenetic tree as the representative and the three *Bemisia tabaci* sHSPs were grouped together in a well-supported cluster (Fig. 2). Furthermore, the three sHSPs of *Bemisia tabaci* also are

**Table 2 Inferred amino acid sequence identities of sHSPs from *Bemisia tabaci* with its homologs from other insects.**

| No. | Species | Name | Accession number |
|-----|---------|------|-----------------|
| 1 | *Bemisia tabaci* | *Bthsp19.5* | AVL92582.1 |
| 2 | | *Bthsp19.2* | AVL92583.1 |
| 3 | | *Bthsp21.3* | AVL92584.1 |
| 4 | | *Bthsp20* B | ACH85196.1 |
| 5 | | *Bthsp20* Q | ADG03464.1 |
| 6 | | *Bthsp20* ZHJ-1 | ADG03467.1 |
| 7 | | *Bthsp20* ZHJ-2 | ADO14472.1 |
| 8 | *Trialeurodes vaporariorum* | *Tvshsp* | ACI15853.1 |
| 9 | | *Tvhsp20* | ACH85200.1 |
| 10 | *Lygus hesperus* | *Lhhsp21.9* | AFX84562.1 |
| 11 | | *Lhhsp21.4* | AFX84561.1 |
| 12 | | *Lhhsp21.5* | AFX84563.1 |
| 13 | *Liriomyza huidobrensis* | *Lhhsp20* | ABE57137.1 |
| 14 | *Musca domestica* | *Mdhsp20* | AHK23446.1 |
| 15 | | *Mdshsp* | ADT92004.1 |
| 16 | *Agasicles hygrophila* | *Ahhsp21* | AHH25011.1 |
| 17 | *Spodoptera litura* | *Slhsp20* | ADK55523.1 |
| 18 | | *Slhsp19.7* | ADK55524.1 |
| 19 | *Bombyx mori* | *Bmhsp20.8* | AAG30944.1 |
| 20 | | *Bmhsp19.9* | BAD74195.1 |
| 21 | | *Bmhsp21.4* | BAD74197.1 |
| 22 | *Locusta migratoria* | *Lmhsp20.5* | ABC84492.1 |
| 23 | | *Lmhsp20.7* | ABC84494.1 |
| 24 | *Oxya chinensis* | *Ochsp20.4* | AJP36907.1 |
| 25 | *Laodelphax striatella* | *Lshsp20.1* | AYP00110.1 |
| 26 | | *Lshsp21.2* | AYP00111.1 |
| 27 | | *Lshsp22.0* | AYP00114.1 |
| 28 | *Chilo suppressalis* | *Cshsp19.7* | BAE94664.1 |
| 29 | | *Cshsp21.4* | AGC23338.1 |
| 30 | | *Cshsp19.8* | AGC23337.1 |
| 31 | *Schistocerca gregaria* | *Sghsp20.7* | AEV89760.1 |
| 32 | *Frankliniella occidentalis* | *Fohsp28.5* | AFX84622.1 |
| 33 | | *Fohsp28.7* | AFX84621.1 |

divided into groups of Hemiptera (*Laodelphax striatella*) and Thysanoptera (*F. occidentalis*) insects, and are far from *Bemisia tabaci* HSP20 which have reported in the past.

### Genomic structure of three genes encoding *Bemisia tabaci* sHSPs

The length of genomic DNA sequences of the three *Bemisia tabaci* sHSPs varied as follows: 1,225 bp for *Bthsp19.5* (GenBank accession no. MF114301); 1,107 bp for *Bthsp19.2* (MF114302); and 1,209 bp for *Bthsp21.3* (MF114303). The cDNA is aligned with the

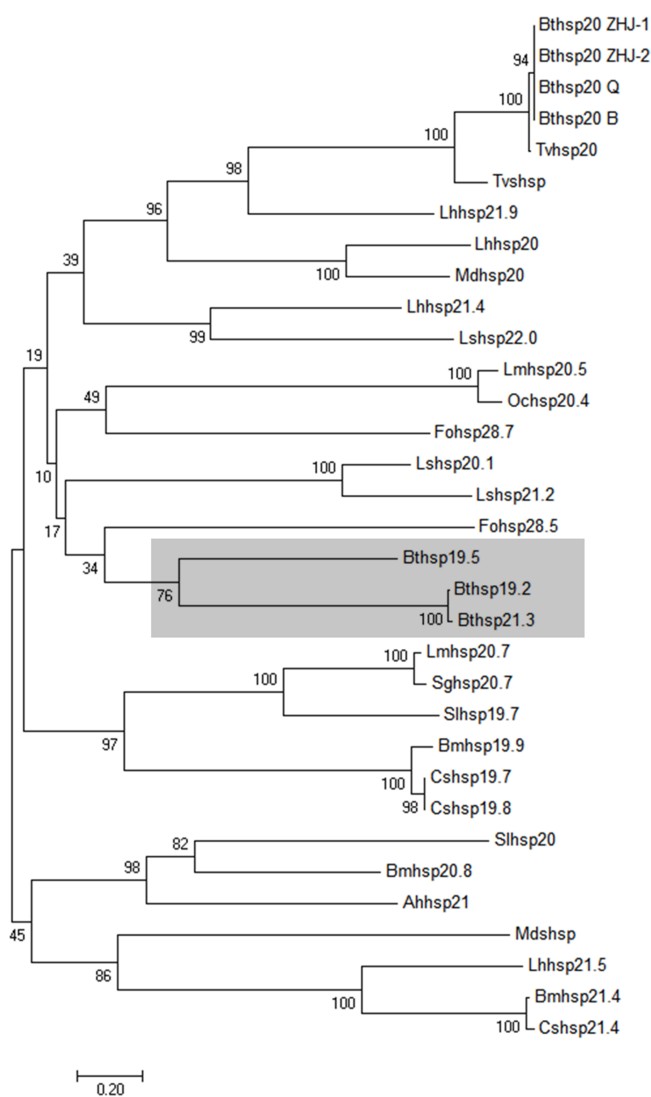

**Figure 2 Neighbor-joining phylogenetic tree of selected insect sHSPs.** Neighbor-joining phylogenetic tree of selected insect sHSPs. Branches containing the *Bemisia tabaci* sHSPs identified in the present study are shaded. Numbers on the branches are bootstrap values obtained from 1,000 replicates (only bootstrap values >50 are shown). Accession numbers and abbreviations for the insect species are listed in Table 2.

genomic DNA sequence and the intron splicing chain nucleotide sequence GT-AG rules were used to find that all the three *Bemisia tabaci* sHSP genes (*Bthsp19.5*, *Bthsp19.2*, and *Bthsp21.3*) lacked introns, which is also the case for *Bmhsp20.4* and *Lmhsp20.7* from *Bombyx mori* and *Locusta migratoria*, respectively (Fig. 3). However, some *shsps* do contain introns; examples include *Cshsp21.4* and *Bmhsp21.4*, which contain one and two introns, respectively (Fig. 3).

## Expression of *Bt*sHSPs in response to temperature stress

The RT-PCR was used to study the expression profiles of the three *shsps* in the MED cryptic species of *Bemisia tabaci*. Melting curve analysis revealed the presence of a single,

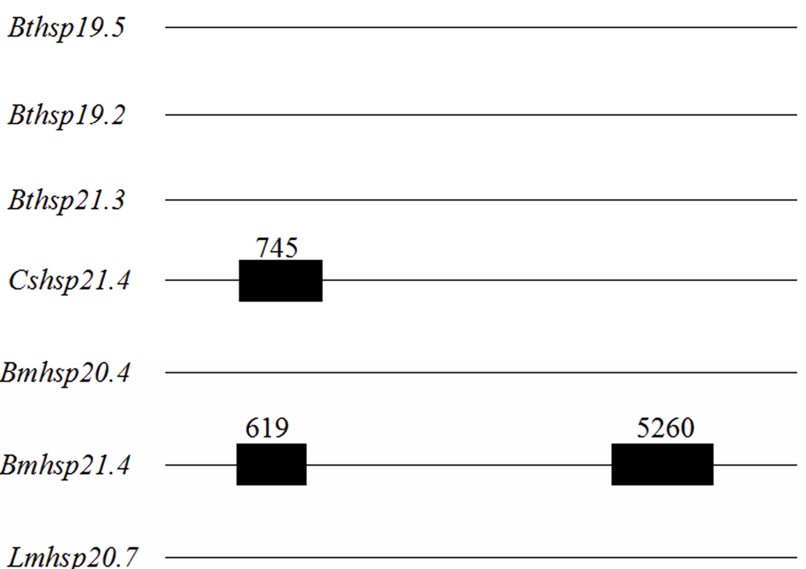

**Figure 3 Schematic representation of *shsps* genomic structure in several insects.** Schematic representation of *shsps* genomic structure in several insects. Species names and accession numbers of genomic DNA sequences are as follows: *Cshsp21.4* (*Chilo suppressalis*, JX491642); *Bmhsp20.4* (*Bombyx mori*, AAG30945); *Bmhsp21.4* (*B. mori*, BAD74197); and *Lmhsp20.7* (*Locusta migratoria*, ABC84494.1). The first three lines represent genomic DNA from the three *B. tabaci shsps* identified in this study (*Bthsp19.5*, *Bthsp19.2*, and *Bthsp21.3*). Horizontal lines and black rectangles are used to demarcate exons and introns, respectively.

sharply defined peak for *Bthsp19.5* ($R^2 = 0.884$), *Bthsp19.2* ($R^2 = 0.938$) and *Bthsp21.3* ($R^2 = 0.992$). The relative mRNA levels of the three *shsps* were compared at −12, −10, −8, −6, 26, 39, 41, 43, and 45 °C. The expression profiles for the three *Bemisia tabaci* sHSP genes were very similar at each of the selected temperatures (Fig. 4). In both adult and pupa forms of *Bemisia tabaci*, *Bthsp19.5* ($F_{17,44} = 223.837$, $P < 0.05$), *Bthsp19.2* ($F_{17,48} = 114.250$, $P < 0.05$) and *Bthsp21.3* ($F_{17,45} = 82.582$, $P < 0.05$) were significantly induced in response to heat stress (39 to 45 °C), whereas expression at cold temperatures (−6 to −12 °C) was not significantly different from the control (26 °C). We also observed differences in *shsps* expression in adult and pupa forms of *Bemisia tabaci* exposed to heat stress. The expression levels of *Bthsp19.2* and *Bthsp21.3* were significantly higher in pupae than adults, and expression was highest at 41 °C (Figs. 4B and 4C). In contrast, the expression of *Bthsp19.5* was greatest in adults exposed to 43 °C (Fig. 4A).

## DISCUSSION

Small heat shock proteins are the least conserved family of HSPs. Although the nucleotide sequence and size of sHSPs are diverse, the majority retain common characteristics (*Franck et al., 2004*; *Haslbeck et al., 2005*). Increasing numbers of studies have found that sHSPs are closely linked to many important functions in insects, such as regulating growth and reproduction, which enhances the ability of insects to adapt to environmental stress, and they are also closely related to insect dormancy and diapause (*Lu, Xu & Du, 2015*).

In this study, we identified three new sHSP genes from the MED cryptic species *Bemisia tabaci*: *Bthsp19.5*, *Bthsp19.2*, and *Bthsp21.3*. The predicted amino acid sequences of

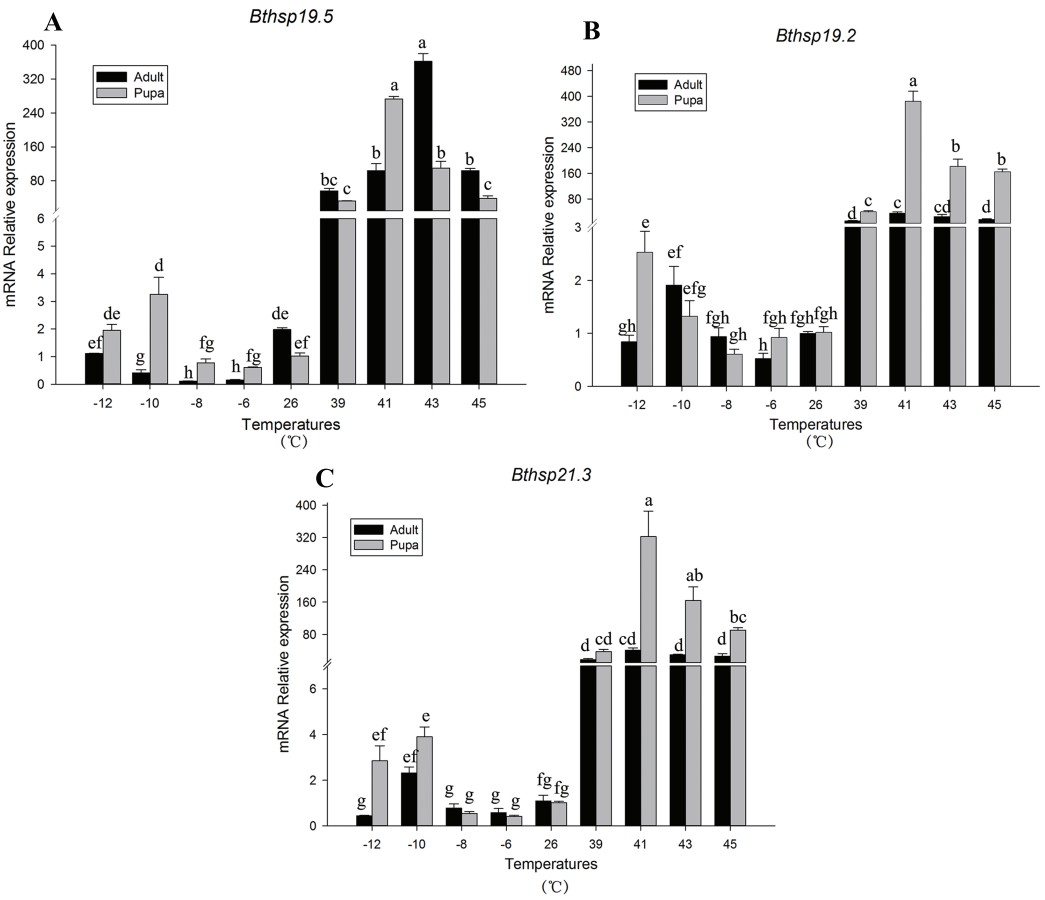

**Figure 4 Relative mRNA expression levels.** Relative mRNA expression levels of (A) *Bthsp19.5*, (B) *Bthsp19.2*, and (C) *Bthsp21.3* in *B. tabaci* pupae and adults exposed to different temperatures. Relative expression levels are shown for *Bthsp19.5*, *Bthsp19.2*, and *Bthsp21.3*. Columns labeled with different letters represent significant differences at $P < 0.05$.

*Bthsp19.5*, *Bthsp19.2*, and *Bthsp21.3* show high sequence similarity to *hsp20* from the MED cryptic species *Bemisia tabaci* and *Trialeurodes vaporariorum*, especially in the α-crystallin protein. Phylogenetic analysis placed the three *Bt*sHSPs proteins in the same cluster (Fig. 3), which comprises a large group containing the Hemiptera species, *Laodelphax striatella*, and Thysanoptera species, *F. occidentalis*. The previously reported *Bemisia tabaci* HSP20 proteins (B, Q, ZHJ-1, ZHJ-2) and HSP20 orthologs from other Hemipteran insects (*T. vaporariorum* and *Lygus hesperus*) grouped separately (*Yu & Wan, 2009*; *Yu, Wan & Guo, 2012a*, *2012b*; *Hull et al., 2013*), indicating that the three *Bt*sHSPs identified in this study belong to the new MED cryptic species of *Bemisia tabaci*. Of course, it remains to be studied further that whether there are other sHSPs in the body of *Bemisia tabaci*.

Previous studies have established a negative conjunction between gene expression and intron size; In other words, genes with short or missing introns are expressed more highly than genes with long or multiple introns (*Comeron, 2004*). Genes of sHSP have been classified into two types: orthogonal and species-specific based on chromosomal location

and intron number (*Li et al., 2009*). Our results indicate that *Bthsp19.5*, *Bthsp19.2*, and *Bthsp21.3* are species-specific forms of sHSP that lack introns.

Prior studies have established that HSPs contribute to temperature tolerance (*Sørensen, Kristensen & Loeschcke, 2003*; *Queitsch, Sangster & Lindquist, 2002*), and our study is no exception. The expression levels of *Bthsp19.5*, *Bthsp19.2*, and *Bthsp21.3* were significantly up-regulated during high temperature stress, indicating that sHSPs function in heat tolerance of the MED cryptic species *Bemisia tabaci*. This result is associated with *Gehring & Wehner (1995)* HSPs involved in multiple physiological processes, one of the most well-known functions is to improve the heat tolerance of insects and protect insects from heat damage and lethal damage. Our findings are consistent with those obtained for *Bthsp20*, which is also induced by heat stress (*Yu, Wan & Guo, 2012a*). Although the role of sHSPs in modulating heat resistance is documented for insects, the response to cold stress is less obvious. For example, *hsp21.4* (*Bombyx mori*), *hsp20*, and *hsp21.4* (*Spodoptera litura*) and *hsp21.4* and *hsp21.7b* (*Chilo suppressalis*) are not sensitive to low temperature stress (*Li et al., 2009*; *Shen et al., 2011*; *Lu et al., 2014*). Similarly, our results indicated that the expression levels of *Bthsp19.5*, *Bthsp19.2*, and *Bthsp21.3* were not modulated by cold shock (Fig. 4). It is important to mention that control of HSP expression is critical in maintaining the cost/benefit ratio of these proteins since over-expression can cause deleterious effects (*Krebs & Feder, 1998*; *Sørensen, Kristensen & Loeschcke, 2003*), further indicating that insects may not have cross resistance between heat and cold adaptation (*Huang, Chen & Kang, 2007*).

Small heat shock proteins also play an important role in insect development (*Concha et al., 2012*; *Shen et al., 2011*; *Takahashi et al., 2010*). For example, the expression of *l(2)efl*, which encodes a sHSP, was highest in third instar larvae of *D. melanogaster* (*Kurzik-Dumke & Lohmann, 1995*). However, in *Lucilia cuprina*, expression of *hsp24* was lowest in third instar larvae (*Concha et al., 2012*). Previous studies on HSPs in *Bemisia tabaci* were primarily focused on adults (*Yu, Wan & Guo, 2012a, 2012b*; *Yu & Wan, 2009*; *Díaz et al., 2015*). In the present study, we observed differential expression of *Btshsps* in adult and pupa forms of *Bemisia tabaci*; for example, expression of *Bthsp19.2* and *Bthsp21.3* was significantly higher in pupae than adults (Figs. 4B and 4C). In contrast, *Bthsp19.5* expression was significantly higher in adults than pupae and showed no significant difference in transcription during high temperature stress. Thus, *Bthsp19.2* and *Bthsp21.3* function during the pupal stage and *Bthsp19.5* expression is important in adults; furthermore, *Bthsp19.5* may have a role in pupae exposed to cold stress (Fig. 4A). Previous studies suggested a link between HSP expression and insect developmental stage. For example, the relative expression levels of three HSP genes of 2nd instar nymphs at each temperature were lower than those of third instar nymphs and female adults in *Phenacoccus solenopsis* (*Chen & Lu, 2014*). In another research, the expression of *Sihsp20.6* and *Sihsp19.6* was highest in eggs of *Sesamia inferens*, whereas the expression of *Sihsp21.4* was highest in adults (*Sun et al., 2014*).

Information on how insect species react to global warming at the physiological and ecological levels is important for predicting epidemiological spread and intrusiveness. Further research is needed to elucidate the role of sHSPs in insect behavior and

development and to determine their relevance to invasiveness. These studies will help to uncover the basic mechanisms of physiological that contribute to insect survival and will improve our capability to carry out more effective control measures finally.

## ACKNOWLEDGEMENTS

We sincerely thank Dr. Carol L. Bender for editing English and helpful comments on the manuscript.

### Funding

This work was supported by the Special Fund for Agro-scientific Research in the Public Interest of China (No. 201303019). The funders had no role in study design, data collection and analysis, decision to publish, or preparation of the manuscript.

### Grant Disclosure

The following grant information was disclosed by the authors:
Special Fund for Agro-scientific Research in the Public Interest of China: 201303019.

### Competing Interests

The authors declare that they have no competing interests.

### Author Contributions

- Jing Bai performed the experiments, analyzed the data, contributed reagents/materials/analysis tools, prepared figures and/or tables, approved the final draft.
- Xiao-Na Liu performed the experiments, analyzed the data, contributed reagents/materials/analysis tools, approved the final draft.
- Ming-Xing Lu conceived and designed the experiments, contributed reagents/materials/analysis tools, authored or reviewed drafts of the paper, approved the final draft.
- Yu-Zhou Du conceived and designed the experiments, authored or reviewed drafts of the paper, approved the final draft.

### DNA Deposition

The following information was supplied regarding the deposition of DNA sequences:
The HSP19.5, HSP19.2, HSP21.3 described here are accessible via GenBank accession numbers MF114301, MF114302 and MF114303.

### Data Availability

The raw measurements are available in the Supplementary Figure.

### Supplemental Information

Supplemental information for this article can be found online at http://dx.doi.org/10.7717/peerj.6992#supplemental-information.

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
