# Peer review of "Characterization of genes encoding small heat shock proteins from Bemisia tabaci and expression under thermal stress"

_PeerJ, doi:10.7717/peerj.6992_

## Round 0.1 · original submission · Minor Revisions

Dear Dr. Bai and colleagues:

Thanks for submitting your manuscript to PeerJ. I have now received three independent reviews of your work, and as you will see, the reviewers raised some minor concerns about the research. Despite this, these reviewers are optimistic about your work and the potential impact it will have on research studying the role of small heat shock proteins in Bemisia tabaci undergoing thermal stress.

Therefore, I am recommending that you revise your manuscript accordingly, taking into account all of the issues raised by the reviewers. I do believe that your manuscript will be ready for publication once these issues are addressed.

Good luck with your revision,

-joe

Reviewer 1 ·

Basic reporting

no comment

Experimental design

no comment

Validity of the findings

no comment

Additional comments

In this paper, the authors isolated and characterized three sHSP family genes. Quantitative Real-time PCR analyses revealed the expression of three BtsHSPs genes under thermal stress. This study gives an important scenario to understand the effect of Bemisia tabaci acclimation. I suggest the manuscript will be accepted for publication after minor revision. Some mistakes have been presented.

In abstract, "In conclusion, our results show that B. tabaci sHSP genes have distinct regulatory roles in the physiology of B. tabaci under thermal stress." — this should be rephrased as "In conclusion, our results show that different B. tabaci sHSP genes show differential expression changes under thermal stress."

Line 57, “study” should be revised to studies.

Line 66, "have replaced by MED" => have been replaced.

Line 68–77, this section of the introduction, which provides the motivation for the present study, is very vaguely phrased ("There are a number of studies on the physiological and biochemical mechanisms of insect stress tolerance … many candidate genes related to temperature adaption and stress have been screened and identified in different insect species"). Please provide a more specific review of what is known about the roles of sHSPs in temperature adaptation in other insect species, since this is the subject of the current study.

Line 87, clarify whether the N numbers (120) are per temperature treatment, or across all temperatures.

Line 91, "Adults and pupae exposed to 26°C were included as a control." — ??: 26°C is one of the treatments (l.95 "exposed to a range of temperatures (-12, -10, -8, -6, 26, 39, 41, 43, and 45°C)") so why an additional 26°C control? Or is this referring to the same 26°C group?

Line 92, "Each treatment included four biological replications." Only here it becomes apparent that N=4, not N=120. N is the statistical sample size; pooling 120 whiteflies in a tube is not "N=120"; the statement on l.94 therefore needs to be rephrased as "cohorts of 120 adults or pupae", and it needs to be made explicit that N=4.

Line 99, an oligo(dT)18 primer 18 should be subscript.

Line 111, "ClustalX software" — needs reference.

Line 116, please cite the reference about MEGA v. 7.0.

Line 154, "The four resulting phylogenetic trees could be divided into two major clusters that were highly similar; thus, only the neighbor-joining tree is shown…" — this makes little sense as written (how can four trees give "two major clusters that were highly similar"?); please rephrase to explain fully.

In the section of Genomic Structure of Three Genes Encoding B. tabaci sHSPs, I found three B. tabaci sHSPs all lack introns, but you analyzed the position and size of the intron of three B. tabaci sHSPs. Please check it.

Line 176, "Melting curve analysis revealed the presence of a single, sharply defined peak…" — please include melting curves as supplemental data. Why was the identity of the qPCR products not confirmed by sequencing?

Line 197, "The previously reported…" needs references.

Reviewer 2 ·

Basic reporting

no comment

Experimental design

no comment

Validity of the findings

no comment

Additional comments

Bai et al. investigated genes encoding small heat shock proteins from Bemisia tabaci and expression under thermal stress. This study is designed well and suitable for publication in this journal. I just have some minor comments:

1. Line 103 the sentence of “Characterization of B. tabaci Genome” should be revised.
2. mRNA relative expression in Figures 4-6 is not correct and should be revised.

Reviewer 3 ·

Basic reporting

Your introduction needs focus on sHSPs of insects and needs more detail.
1 I suggest that you improve the description at lines 54-58. There are a lot of works with agricultural insects,multiple small heat-shock protein genes have been identified in Plutella xylostella, Chilo suppressalis, Dialeurodes citri, Spodoptera litura.
2 The sentence you use in Lines 64-71 is not closely related to the subject.
3 Temperatures in Figures 4-6 should labeled with ℃,and Figure 3 is not necessary.
4 Only 3 sHSPs genes were isolated from Bemisia tabaci, which is less than other insects. I suggest the authors explained it in part Discussion.

Experimental design

no comment

Validity of the findings

no comment

---

## Round 0.2 · accepted · Accept

Dear Dr. Bai and colleagues:

Thanks for revising your manuscript based on the minor concerns raised by the reviewers. I now believe that your manuscript is suitable for publication. Congratulations! I look forward to seeing this work in print, and I anticipate it being an important resource for research communities studying the role of small heat shock proteins in Bemisia tabaci undergoing thermal stress. Thanks again for choosing PeerJ to publish such important work.

Best,

-joe

#